# A large meta-analysis identifies genes associated with anterior uveitis

Sahar Gelfman[1], Arden Moscati[1], Santiago Mendez Huergo[2], Rujin Wang[1], Veera Rajagopal[1], Neelroop Parikshak [1], Vijay Kumar Pounraja [1], Esteban Chen[1], Michelle Leblanc[1], Ralph Hazlewood [2], Jan Freudenberg[1], Blerta Cooper[2], Ann J. Ligocki[2], Charles G. Miller[2], Tavé Van Zyl[2], Jonathan Weyne[2], Carmelo Romano[2], Botir Sagdullaev[2], Olle Melander[3], Aris Baras [1], Regeneron Genetics Center*, Eli A. Stahl [1,4] ✉ & Giovanni Coppola [1,4] ✉

Anterior Uveitis (AU) is the inflammation of the anterior part of the eye, the iris and ciliary body and is strongly associated with *HLA-B*27*. We report AU exome sequencing results from eight independent cohorts consisting of 3,850 cases and 916,549 controls. We identify common genome-wide significant loci in *HLA-B* (OR = 3.37, *p* = 1.03e-196) and *ERAP1* (OR = 0.86, *p* = 1.1e-08), and find *IPMK* (OR = 9.4, *p* = 4.42e-09) and *IDO2* (OR = 3.61, *p* = 6.16e-08) as genome-wide significant genes based on the burden of rare coding variants. Dividing the cohort into *HLA-B*27* positive and negative individuals, we find *ERAP1* haplotype is strongly protective only for B*27-positive AU (OR = 0.73, *p* = 5.2e-10). Investigation of B*27-negative AU identifies a common signal near *HLA-DPB1* (rs3117230, OR = 1.26, *p* = 2.7e-08), risk genes *IPMK* and *IDO2*, and several additional candidate risk genes, including *ADGFR5*, *STXBP2*, and *ACHE*. Taken together, we decipher the genetics underlying B*27-positive and -negative AU and identify rare and common genetic signals for both subtypes of disease.

Uveitis is an intraocular inflammatory disease which can result in severe visual loss[1–3] and can be categorized by etiology (infectious or non-infectious) and by affected ocular region (anterior, intermediate, posterior, or panuveitis). Non-infectious uveitis represents the majority of cases in the developed world, with a prevalence of 121 per 100,000 in the US[3,4]. Anterior Uveitis (AU), characterized by inflammation of the iris and/or ciliary body, is the most common type of non-infectious uveitis, with a prevalence of 98/100,000 adults in the US, accounting for ~80% of all non-infectious uveitis cases[3]. AU predominately affects younger individuals, with a mean age of onset less than 40 years of age[5,6].

AU is frequently observed as a complication of spondyloarthropathies (SpAs), such as ankylosing spondylitis (AS), psoriatic arthritis, and inflammatory bowel disease (IBD)[7]. Hence, most studies so far have focused on AU in the context of a spondyloarthropathy (mainly AS). Notably, all these inflammatory diseases are strongly associated with HLA haplotypes. Birdshot Chorioretinopathy has the strongest known association to human leukocyte antigen (HLA)-A*29, followed by AS and *HLA-B*27*[8]. Such an association was first described for AS and AU 50 years ago[9,10], and since confirmed in several studies[11–14]. It has previously been estimated that approximately 50% of all patients with AU are *HLA-B*27* positive[15], which increases to above 80% among AS patients with AU[11]. In addition to *HLA-B*27*, associations of smaller effect size were described with other HLA alleles including *HLA-A*02:01, HLA-B*08, HLA-DRB1*15 and HLA-DPB1*03*[14], and with other non-HLA common loci, including *ERAP1, IL23R*, and the 2p15 locus[16]. To date, most genetic association studies of AU were mostly in the setting of AS, with the drawback of intertwining the genetic signals of both diseases.

[1]Regeneron Genetics Center, 777 Old Saw Mill River Rd., Tarrytown, NY 10591, USA. [2]Regeneron Pharmaceuticals, 777 Old Saw Mill River Rd., Tarrytown, NY 10591, USA. [3]Department of Clinical Sciences Malmö, Lund University, 221 00 Malmö, Sweden. [4]These authors contributed equally: Eli A. Stahl, Giovanni Coppola. *A list of authors and their affiliations appears at the end of the paper. ✉e-mail: eli.stahl@regeneron.com; giovanni.coppola@regeneron.com

Here, we evaluate several large AU cohorts consisting of 3850 cases in total and match them with over 900,000 controls in the largest AU meta-analysis to date. We describe the underlying genetics of B*27-positive (B*27-pos) and B*27-negative (B*27-neg) AU in the various ancestries to identify strong signals in both sub-types. A B*27-pos analysis identifies a significant, *HLA-B*27*-dependent protective signal in *ERAP1*, suggesting an altered immunogenic peptidome as a pathogenetic factor. A complementary analysis of 2984 B*27-neg AU cases identifies both common and rare signals for B*27-neg AU: a genome-wide significant common signal near the HLA Class-II *HLA-DPB1* gene, and several genome-wide significant genes that increase the risk for AU identified through gene-burden analyses of rare damaging coding variants. These results shed light on the genetics of AU and stress the importance of whole-exome sequencing in the efforts to decipher the disease's underlying genetic risks.

## Results

### Common genetic signals contributing to AU risk

We sequenced eight large EHR based populations, including 3850 AU cases and 916,549 controls (Table 1). Testing the association of common variants, we discovered two genome wide significant signals for AU: a risk signal at the *HLA-B* locus (rs543685299, OR [95% CI] = 3.37 [3.11−3.65], $p = 1.03E−196$) and a protective signal for rs3198304 at the *ERAP1* locus (OR [95% CI] = 0.86 [0.82−0.91], $p = 1.1e−8$) (Fig. 1). The top *ERAP1* SNP showed a consistent direction of effect in 6/8 cohorts (Fig. 2).

We repeated the analysis while restricting to individuals of European descent (3,180 cases and 826,685 controls) and observed similar results for both *HLA-B* (OR = 3.4, $p = 1.1e−185$) and *ERAP1* (OR = 0.85, $p = 1.1e−08$, Fig. S1−S2). *ERAP1* is an ER-aminopeptidase that trims peptides to be loaded and presented by MHC class-I proteins, and alterations in *ERAP1* change the peptidome available to HLA Class I alleles[17].

### Rare variant analyses identify risk genes contributing to AU risk

We next tested several gene-burden models that incorporated various AF filter thresholds as well as variant deleteriousness scores (see Methods).

The gene burden analyses combining all cohorts exhibited a controlled low inflation of $\lambda=0.94$ (Fig. S3), suggesting that the analysis did not deviate from the expected p value distribution and was well-adjusted for population stratification. This allowed us to confidently identify genes that pass a strict study-wide and genome-wide significance threshold. We used a study-wide significance threshold of $p = 2.86e−07$ calculated by using the approach from Li & Ji 2005 for multiple testing correction[18], and utilized this for the remainder of results discussed here (see Methods for details).

Five genes reached the study-wide significance threshold ($p = 2.86e−07$, Methods). The first, *IPMK*, was significant when considering a model that includes pLoF and missense variants that are strongly deleterious (predicted by 5/5 prediction models), with AF < 0.1%, reaching a high OR [95% CI] = 9.42 [4.44−19.89] with $p = 4.4e−09$ (Table 2, Supplementary Data 1−2, Supplementary Information).

The second genome-wide and study-wide significant gene, *IDO2*, showed a strong risk signal with OR [95% CI] = 3.61 [2.23-5.7], $p = 6.16e−08$ for rare (AF < 0.1%) pLoF variants. *IDO2* is a LoF-tolerant gene that exhibits a pLI score of 0 and O/E = 0.81 (0.54−1.25)[19]. The top association burden included seven distinct pLoF variants (Supplementary Data 2−3, Supplementary Information).

Three additional genes exhibited borderline significant $p$ values and represented results from extremely rare gene-burden masks that consider only variants appearing once in each cohort (singletons). The first gene, *ACHE*, had six cases, each carrying an extremely rare and distinct damaging missense variant, with OR [95% CI] = 15.29 [5.57-42.0], $p = 1.22e−07$. Since no pLoF variants are included in this model, this might suggest a gain of toxic function to this gene that might affect AU risk. *ACHE* codes for acetylcholinesterase, a well-known enzyme that breaks down acetylcholine (Ach). *STXBP2* (Syntaxin binding protein 2) was also significant when considering an extremely rare, missense-only gene burden mask, with nine cases carrying distinct variants (OR [95% CI] = 11.66 [4.63-29.39], $p = 1.92e−07$). Missense and PLoF mutations in *STXBP2* are associated with autosomal-recessive Familial Hemophagocytic Lymphohistiocytosis (FHL), a hyperinflammatory syndrome caused by uncontrolled overactivation of the immune response. Uveitis has been reported as a manifestation of hemophagocytic lymphohistiocytosis[20,21]. Lastly, five extremely rare pLoF variants in *ADGRF5*, the adhesion G-protein coupled receptor 5 (also called *GPR116*), aggregate to increase risk for AU (OR [95% CI] = 27.04 [7.73-94.54], $p = 2.44e−07$). The low number of case carriers observed for *ACHE*, *STXBP2*, and *ADGRF5* (<10), suggests that further support is required to nominate them as risk genes for AU (Supplementary Information).

### *ERAP1* signal is strengthened in a B*27-stratified analysis

To better understand the genetic signals underlying *HLA-B*27* in the AU cohort, we next controlled for *HLA-B*27* in our analyses using the *HLA-B*27*-tagging SNP rs4349859 as a covariate. When controlling for this SNP, the signal at the HLA locus was diminished, leaving a borderline signal near *HLA-DPB1* (OR = 1.15, $p = 6e−08$, Fig. S4). However, the protective signal on *ERAP1* remained genome-wide significant (OR = 0.84, $p = 1.7e−8$). Thus, after conditioning for the *HLA-B*27* signal, we still observed associations at both the *ERAP1* and HLA loci. We

**Table 1 | Overview of eight cohorts included in the meta-analysis**

| Cohort | EUR | | ALL | |
|---|---|---|---|---|
| | Cases (HLA-B*27; %) | Controls (HLA-B*27; %) | Cases (HLA-B*27; %) | Controls (HLA-B*27; %) |
| UKB | 1260 (420; 33.5%) | 429,728 (35,066; 8.2%) | 1388 (424; 30.7%) | 452,976 (35,464; 7.9%) |
| GHS | 1007 (184; 18.3%) | 150,775 (12,216; 8.1%) | 1066 (188; 17.6%) | 159,644 (12,481; 7.8%) |
| UPenn-PMBB | 75 (13; 17.6%) | 28,426 (2133; 7.5%) | 233 (19; 8.2%) | 41,304 (2,352; 5.7%) |
| Sinai | 51 (13; 25.5%) | 10,681 (603; 5.7%) | 169 (20; 11.9%) | 29,676 (1,088; 3.7%) |
| MALMO | 114 (30; 26.3%) | 28,834 (2807; 9.7%) | 116 (30; 25.9%) | 29,121 (2,818; 9.7%) |
| MAYO | 331 (72; 21.8%) | 110,930 (9711; 8.8%) | 355 (74; 20.8%) | 115,063 (9,902; 8.6%) |
| UCLA | 161 (36; 22.4%) | 26,276 (1880; 7.2%) | 305 (54; 17.7%) | 39,912 (2,306; 5.8%) |
| Colorado | 181 (40; 22.1%) | 41,035 (3356; 8.2%) | 218 (47; 21.6%) | 48,853 (3,698; 7.6%) |
| Total | 3180 (808; 25.5%) | 826,685 (67,772; 8.2%) | 3850 (856; 22.3%) | 916,549 (70,109; 7.7%) |

Calculations of *HLA-B*27* percentage of carriers include both cases and controls, but exclude samples with missing *HLA-B*27* information (detailed in Supplementary Data 9). *UKB* U.K. Biobank, *GHS* Geisinger Health System MyCode, *UPenn-PMBB* The Penn Medicine BioBank, *Sinai* Mount Sinai BioMe BioBank, *MALMO* Malmö Diet and Cancer Study, *Colorado* Colorado Center for Personalized Medicine Biobank, *UCLA* UCLA ATLAS Community Health Initiative Biobank, *MAYO* MAYO-Clinic RGC Project Generation.

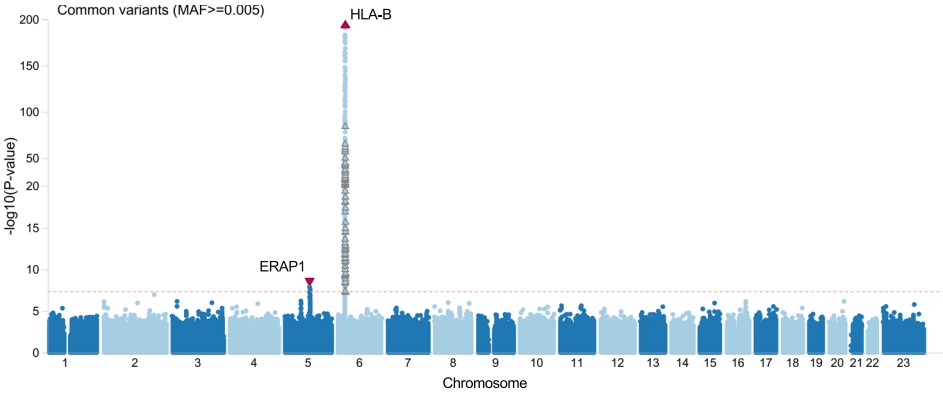

**Fig. 1 | Common *HLA-B* risk and *ERAP1* protection with 3850 AU cases and 916,549 controls.** A Manhattan plot depicting the -log10(*P* value) for all common variants (y-axis) across all chromosomes (x-axis). *HLA-B* top risk signal is shown by an upward red triangle on chromosome six, while *ERAP1* protection is shown by the downward red triangle on chromosome five. Association models were run with age, age$^2$, sex and age × sex, and 10 ancestry-informative principal components as covariates. *P* values are uncorrected and are from two-sided tests performed using approximate Firth logistic regression.

therefore designed stratified analyses by which we divided the cohorts by the carrier status of *HLA-B\*27* using the *HLA-B\*27* tag SNP rs4349859. The *HLA-B\*27* stratification resulted in two cohorts: (1) a B\*27-pos cohort with samples carrying either one or two copies of the tag SNP, and (2) a B\*27-neg cohort with samples carrying zero copies of the tag SNP. The B\*27-pos cohort consisted of 856 AU cases and 70,109 controls, suggesting that 22.2% of the analyzed AU cases carry the B\*27 allele. This is a significant enrichment compared to the 7.7% B\*27 carriers in the controls, similar to the 6%-8% expected *HLA-B\*27* frequency in general population in the US[22].

The final B\*27-stratified analysis greatly weakened the *HLA-B* signal (rs543685299: OR = 1.49; *p* = 5.6e-3), while the *ERAP1* signal remained the only genome-wide significant locus (Fig. 3A, B). Moreover, the protective effect of the *ERAP1* variant was stronger when examining the smaller B\*27-stratified cohort (rs27710, OR = 0.74, *p* = 1.3e-9), even though the B\*27-pos cohort included only 22.2% of cases and 7.7% of controls from the larger cohort.

The top *ERAP1* variant is in perfect LD with the *ERAP1* missense variant K528R-rs30187 (Fig. 3B, R$^2$ = 1, D' = 1). This haplotype has been shown to be an eQTL that significantly decreases *ERAP1* expression and is associated with other HLA class-I related disorders such as AS and Spondyloarthritis[23]. The effect of the *ERAP1* signal across the B\*27-pos cohorts was consistently protective (Fig. 3C). We repeated the analysis while restricting to individuals of European descent consisting of 808 AU cases and 67,761 controls. The EUR-only, B\*27-pos analysis confirmed the significant results for the protective *ERAP1* locus with OR [95% CI] = 0.73 [0.66–0.81], and a *p* = 4.1e-10 for the top SNP rs30187 (Fig. S5).

Since AU is commonly observed in other class-I-opathies such as Psoriatic Arthritis and AS, we designed a strict analysis removing all samples diagnosed with either AS (ICD10-M45) or psoriasis (ICD10-L40) from the smaller B\*27-pos cohort. When considering only B\*27 carriers that were not diagnosed with either AS or psoriasis, we identified 618 AU cases and 67,256 controls in all eight cohorts including all ancestries. This sets the proportion of AU cases that are diagnosed also with AS or Ps at 28%. Within the general B\*27 controls, we found 4% to have AS or Ps diagnosis. In this analysis, *ERAP1* locus presented a similar protection of OR = 0.74, with *p* = 3e-6 owing to the decreased power of this analysis, but supporting the protective direction of the full analysis (Fg. S6).

**Phasing of the *ERAP1* locus identifies the risk and protection *ERAP1* haplotypes**

*ERAP1* haplotypes were previously studied in the context of several HLA class I-associated autoimmune diseases including Birdshot

Chorioretinopathy (BSCR) and AS[24–26]. The common haplotypes are reported to affect *ERAP1* expression levels and enzymatic activity. Haplotypes Hap2 and Hap3 associate with increased expression and enzymatic activity, while Hap10 corresponds to a decrease in both expression and activity[17,25]. The main SNPs that distinguish between these sets of haplotypes, K528R (rs30187) and D575N (rs10050860) show a distinct eQTL effect on *ERAP1* expression as observed in the GTEx data for many tissues, the strongest including whole blood (*p* = 4.4e-78), skeletal muscle (*p* = 4.6e-49) and lung (*p* = 6.1e-46) for rs30187, and skeletal muscle (*p* = 5.0e-43), whole blood (*p* = 9.8e-31) and esophagus (*p* = 1.0e-14) for rs10050860[27]. We therefore set out to examine all possible *ERAP1* haplotypes and their effect on AU risk.

The phasing of *ERAP1* common SNPs that construct the *ERAP1* haplotypes included: (a) the extraction of the distinctive imputed *ERAP1* SNPs, (b) phasing the dosage data, and (c) classifying individual SNPs in each sample into one of the 10 defined haplotypes (described in the Methods section). We then modeled the association of each haplotype with case-control status, including the covariates of age and sex. The results pointed to Hap2 as the top risk for AU with OR = 1.2 and *p* = 2.1e-09 (Table 3). Interestingly, Hap1, which differs from Hap2 only by variants I12 (rs72773968) and 127 P (rs26653) and occurs at a similar frequency, was not significant, suggesting that residues 12 and 127 contribute to the Hap2 risk. When tested individually, 12 T has OR = 1.13 (*p* = 6.8e-04) and R127 has OR = 1.08 (*p* = 2.7e-03) in the most-powered variant level association including the full cohort.

The results were most pronounced for Hap10, which presented a strong protective signal (OR = 0.83, *p* = 2.8e-10) driving most of the *ERAP1* protective signal observed in the previous analyses. Hap10 represents the strongest common eQTLs that decrease the expression of *ERAP1* including 528 R and 575 N. We also identified a protective effect for Hap6 (OR = 0.85; *p* = 2e-04) that shares most SNPs with Hap10 including 528 R and 730E, with the two SNPs having strong effects on decreased *ERAP1* expression and activity[27,28].

We applied the same approach but to the smaller B\*27-pos cohort. We hypothesized that since *ERAP1* signal is specific to this HLA allele background, the haplotype effects will become more prominent. The analyses confirmed this hypothesis, presenting the same risk direction for Hap2 (OR = 1.38, *p* = 5.38e-07) and protection for Hap10 (OR = 0.71, *p* = 4.4e-07), exhibiting stronger effects and weaker *p* values due to the significant loss of power (Supplementary Data 4). The protective effect of Hap6 is also more prominent in the B\*27-pos cohort, with a strong protective OR = 0.7 and similar *p* = 4e-04, surprisingly maintaining the same signal with the much smaller cohort, due to the stronger depletion in cases.

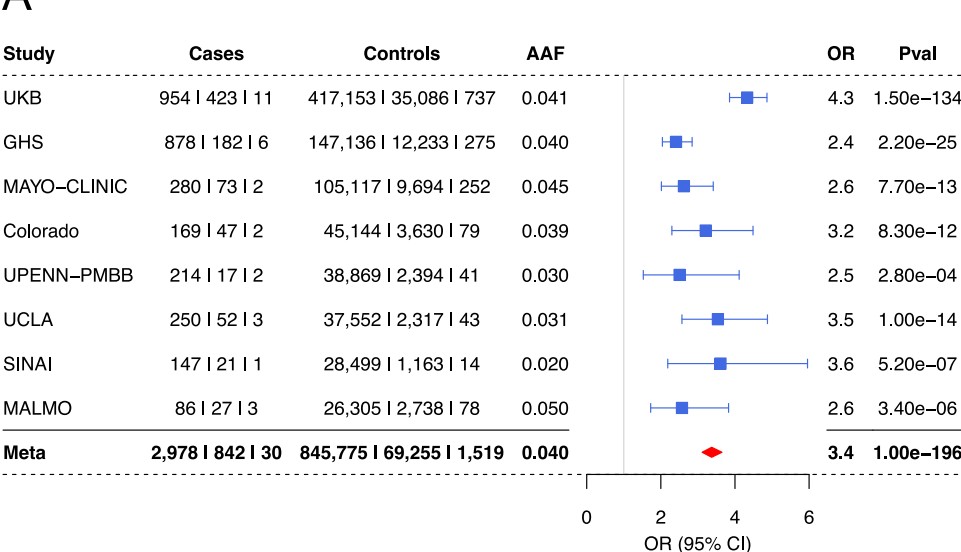

**Fig. 2 | Top SNPs at the *HLA-B* and *ERAP1* loci across eight cohorts. A** A forest plot depicting the association details for *HLA-B* top risk variant rs543685299 in each of the eight cohorts tested and including all ancestries. **B** A forest plot depicting the association details for the top *ERAP1* protective intronic variant rs3198304 in the eight cohorts tested and including all ancestries. A meta-analysis result combining all cohorts is the lowest row (bold), meta-analysis OR is presented by a red diamond. Center points represent odds ratios as estimated by approximate Firth logistic regression, with errors bars representing 95% confidence intervals. *P* values are uncorrected and reflect two-sided tests. Numbers below the cases and controls columns represent counts of individuals with homozygote reference, heterozygote and homozygous alternative genotypes, respectively.

**Table 2 | Top gene burden results for AU**

| Gene | Burden mask | AF | OR | Cases (ref \| het \| alt) | Controls (ref \| het \| alt) | Pval |
|---|---|---|---|---|---|---|
| *IPMK* | pLoF and damaging missense (5/5) | <0.1% | 9.40 | 3839 \| 11 \| 0 | 916,138 \| 411 \| 0 | 4.42E−09 |
| *IDO2* | pLoF only | <0.1% | 3.61 | 3822 \| 27 \| 1 | 915,111 \| 1433 \| 5 | 6.16E−08 |
| *ACHE* | damaging missense (5/5) | singleton | 15.29 | 3844 \| 6 \| 0 | 916,388 \| 161 \| 0 | 1.22E−07 |
| *STXBP2* | damaging missense (5/5) | singleton | 11.66 | 3841 \| 9 \| 0 | 916,269 \| 280 \| 0 | 1.92E−07 |
| *ADGRF5* | pLoF only | singleton | 27.04 | 3845 \| 5 \| 0 | 916,405 \| 144 \| 0 | 2.44E−07 |

Association models were run with age, age², sex and age × sex, and 10 ancestry-informative principal components as covariates. *P* values are uncorrected and are from two-sided tests performed using approximate Firth logistic regression. Numbers below the cases and controls columns represent counts of individuals with homozygote reference, heterozygote and homozygous alternative genotypes, respectively.

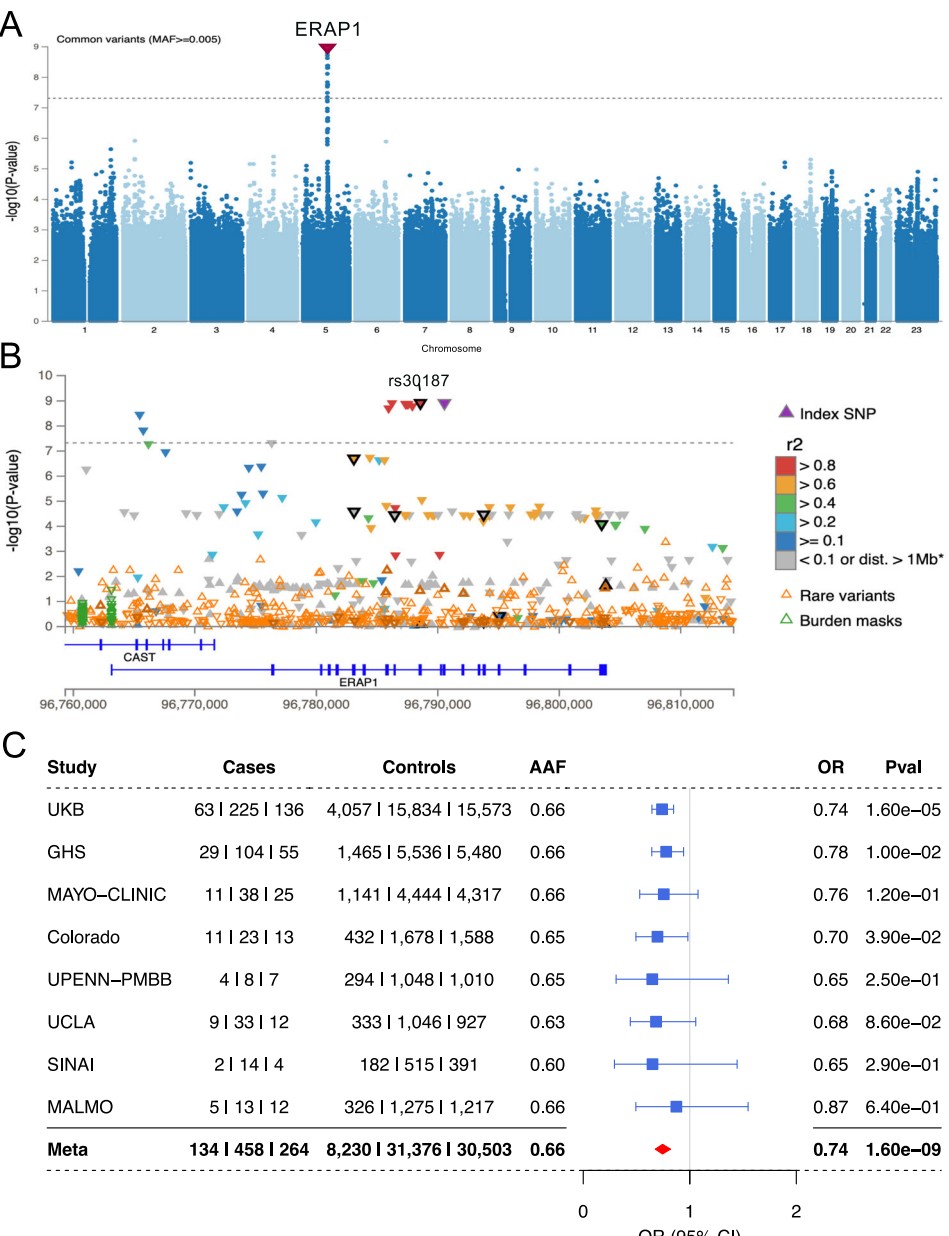

**Fig. 3 | A B\*27-pos analysis exhibiting *ERAP1* as the only genome-wide significant risk for B\*27-AU.** **A** A Manhattan plot depicting the -log10(*P* value) for all common variants (y-axis) across all chromosomes (x-axis). *ERAP1* top protective signal is shown by a downward red triangle on chromosome five. **B** A locus zoom plot showing *ERAP1*. Genome-wide significant threshold of 5e−08 is represented by a dashed gray line. Coding variants are highlighted in black, including labeled rs30187 (K528R). **C** A forest plot depicting the association details for *ERAP1* top risk variant (rs30187) in all cohorts tested. A meta-analysis result combining all cohorts is the lowest row (bold), meta-analysis OR is represented by a red diamond. Center points represent odds ratios as estimated by approximate Firth logistic regression, with errors bars representing 95% confidence intervals. *P* values are uncorrected and reflect two-sided tests. Numbers below the cases and controls columns represent counts of individuals with homozygote reference, heterozygote and homozygous alternative genotypes, respectively.

## An additive effect for B\*27-AU risk with the combined effect of having two copies of HLA risk alleles and the *ERAP1* risk haplotypes

The effects of the *ERAP1* risk-Hap2 and protective-Hap10 were assessed in the above analysis in all subjects or subjects carrying at least one copy of the *HLA-B\*27* allele. We next constructed a model to test the effect of homozygous/heterozygous *ERAP1* haplotypes on different *HLA-B\*27* backgrounds, using samples that carry no B\*27 risk alleles, one, or two copies of the B\*27 risk-allele. We defined zero *HLA-B* and *ERAP1* protection haplotypes (two copies Hap10 and no copies of Hap2) as the reference risk genotype (i.e. OR = 1), and assessed the risk of the *ERAP1* Hap10 and Hap2 on a B\*27 negative background (Fig. 4, left panel and supplementary Data 5), compared to having one (Fig. 4,

middle panel) and two copies of *HLA-B\*27* allele (Fig. 4, right panel). We found a moderate risk increase of OR = 1.4 for individuals carrying two protective *ERAP1*-Hap10 copies with one copy of *HLA-B\*27* allele, which increased by more than four times (OR = 6.3) when replacing two copies of *ERAP1*-Hap10 with two copies of *ERAP1*-Hap2. The maximum risk combination (two *ERAP1*-Hap2 and two *HLA-B\*27* alleles), reached a large OR = 36.9. We found that even with two copies *HLA-B\*27*, having two copies of *ERAP1*-Hap10 reduces the AU risk back to a model-estimated OR = 2.1. However, we did not observe cases carrying two copies of Hap10 and two copies of *HLA-B\*27*, suggesting that the risk for AU when having two copies of *ERAP1* Hap10 is even lower. This result supports the hypothesis that the *ERAP1*-Hap2 (increased activity and expression of *ERAP1*) play a role in the processing of the antigenic

**Table 3 | ERAP1 haplotype associations with AU**

| Haplotype | ERAP1 Haplotype-defining variants | | | | | | | | | | Effects and descriptive statistics in all individuals | | | | | |
|---|---|---|---|---|---|---|---|---|---|---|---|---|---|---|---|---|
| | rs72773968 5:96803892:G-A I12T | rs37340016 5:96803761:C-T E56K | rs26653 5:96803547:C-G R127P | rs26618 5:96795133:T-C I276M | rs27895 5:96793840:C-T G346D | rs2287987 5:96793832:T-C M349V | rs30187 5:96788627:T-C K528R | rs10050860 5:96786506:C-T D575N | rs17482078 5:96783162:C-T R725Q | rs27044 5:96783148:G-C Q730E | Frequency in Cases | Frequency in Controls | Case carriers (none \| Het \| Hom) | Control carriers (none \| Het \| Hom) | OR | P value |
| Hap1 | I | E | P | I | G | M | K | D | R | Q | 0.124 | 0.12 | 2955\|827\|68 | 709557\|192846\|13922 | 1.04 | 0.29 |
| Hap2 | T | E | R | I | G | M | K | D | R | Q | 0.17 | 0.15 | 2631\|1100\|119 | 664597\|229109\|22619 | 1.2 | 2.1E–09 |
| Hap3 | T | E | R | I | G | M | K | D | R | E | 0.078 | 0.075 | 3264\|559\|27 | 784510\|126457\|5358 | 1.08 | 0.083 |
| Hap4 | T | E | R | I | G | M | R | D | R | E | 0.011 | 0.009 | 3764\|83\|3 | 899724\|16393\|208 | 1.27 | 0.023 |
| Hap5 | T | E | R | I | D | M | R | D | R | E | 0.065 | 0.067 | 3370\|457\|23 | 798796\|113267\|4262 | 0.98 | 0.7 |
| Hap6 | T | E | P | I | G | M | R | D | R | E | 0.073 | 0.085 | 3307\|518\|25 | 766984\|142426\|6915 | 0.85 | 2.0E–04 |
| Hap7 | T | K | P | I | G | M | R | D | R | E | 0.045 | 0.043 | 3532\|306\|12 | 839129\|75186\|2010 | 1.0 | 0.96 |
| Hap8 | T | E | P | M | G | M | R | D | R | E | 0.227 | 0.229 | 2317\|1326\|207 | 544518\|323330\|48477 | 0.98 | 0.42 |
| Hap9 | T | E | P | M | G | M | R | N | R | E | 0.001 | 0.002 | 3843\|7\|0 | 913310\|3012\|3 | 0.55 | 0.12 |
| Hap10 | I | E | P | I | G | V | R | N | Q | E | 0.174 | 0.199 | 2655\|1080\|115 | 590085\|288170\|38070 | 0.83 | 2.8E–10 |

Association models were run with age, age², sex and age × sex as covariates. P values are uncorrected and are from two-sided tests performed using binomial logistic regression.

peptide(s) that is presented by *HLA-B\*27* in AU. The Hap10 haplotype, that is associated with decreased activity and expression, might process a peptidome that lacks the antigenic peptide(s).

### HLA-DPB1 is a significant risk for B\*27-neg AU

We next included only cases and controls not carrying the B\*27-tagging SNP (B\*27-neg AU). This cohort consisted of 2984 B\*27-neg AU cases and 844,709 B\*27-neg controls. This analysis revealed a genome-wide significant signal at rs6914651, an HLA-Class-II gene region near *HLA-DPB1*, which had not previously been associated with AU [OR = 1.18 (1.11–1.25), $p = 1.6e{-}08$] (Fig. 5). With an allele frequency of 0.277, rs6914651 tags a common signal that might reflect a coding variant within *HLA-DPB1* that is associated with AU risk, which in turn might point to a specific *HLA-DPB1* allele that increases risk of *HLA-B\*27* negative AU. To answer this question, we followed with two additional analyses: (1) imputing the *HLA-DPB1* alleles and testing for association of each allele with case-control status, and (2) fine-mapping of the region near *HLA-DPB1* to uncover the genetic signals that underlie this significant association. The results of testing the associations of class-II HLA alleles have shown *HLA-DPB1\*04:01* as a protective allele ($p = 7.2e{-}06$, OR = 0.89) and *HLA-DPB1\*03:01* as risk ($p = 2.4e{-}04$, OR = 1.2, supplementary Data 6). However, when adjusting for the top SNP (rs6914651) in the regression model, neither *HLA-DPB1\*03:01* or *HLA-DPB1\*04:01* were nominally significant, suggesting that it might not be a specific *HLA-DPB1* allele that affects AU risk (supplementary Data 7). However, rs6914651 acts as an eQTL for *HLA-DPB1* and significantly decreases its expression, supporting an effect on AU risk by decreasing *HLA-DPB1* expression[27]. The results of fine-mapping the DPB1 region also suggested that the signal originates not from HLA-DPB1 itself, but from the region downstream to *HLA-DPB1*, where a long stretch of non-coding variants share similar posterior inclusion probabilities (Fig. S7).

### Gene burden analyses of B\*27-neg AU

We next asked whether the gene burden analyses using the B\*27-neg AU cohort replicate previous results with the full cohort. This question is highly relevant to deciphering the mechanism underlying both subtypes of AU. We found that both *IPMK* and *IDO2* replicated a similar direction of risk in the B\*27-neg cohort (Table 4, Supplementary Information). Both of those genes also did not show significant associations in the B\*27-pos cohort, suggesting these mechanisms of risk pertain to B\*27-neg AU.

Aside from *IPMK* and *IDO2*, we found support for several additional genes when examining only the B\*27-neg cohort (Supplementary Data 8). First, the signal for *ADGRF5* has an OR of [95% CI] = 27.6 [10.1–75.5] and $p = 1e{-}10$, due to the addition of two B\*27-neg cases carrying singleton pLoF variants. While the number of cases is still low (<10), this gives additional support for *ADGRF5* to be involved in risk of AU. In addition, the same *STXBP2* model including singleton damaging missense variants was also strengthened to OR[95% CI] = 14.8 [6.1–35.8] and $p = 2.3e{-}09$ with the addition of one case. As *ADGRF5*, this analysis also provided additional support for *STXBP2* to be involved in the risk for AU. We further identified PMP22 as a borderline gene with to OR[95% CI] = 4.88 [2.7–8.9] and $p = 2.3e{-}07$, with a mask that includes rare missense variants and AF < 0.01%. Last, two additional genes received a borderline $p$ value below threshold with six carriers each, for rare pLoF and missense singleton masks, respectively: *LDHA* (OR[95% CI] = 16.6 [11.6–218.5], $p = 4.7e{-}08$) and *DPH6* (OR[95% CI] = 16.3 [5.6–47.1], $p = 2.8e{-}07$). However, lacking additional support, these candidate genes will require further evidence to be considered as AU risk.

## Discussion

Anterior uveitis (AU) is often studied as a manifestation of systemic autoimmune diseases, with high prevalence in seronegative

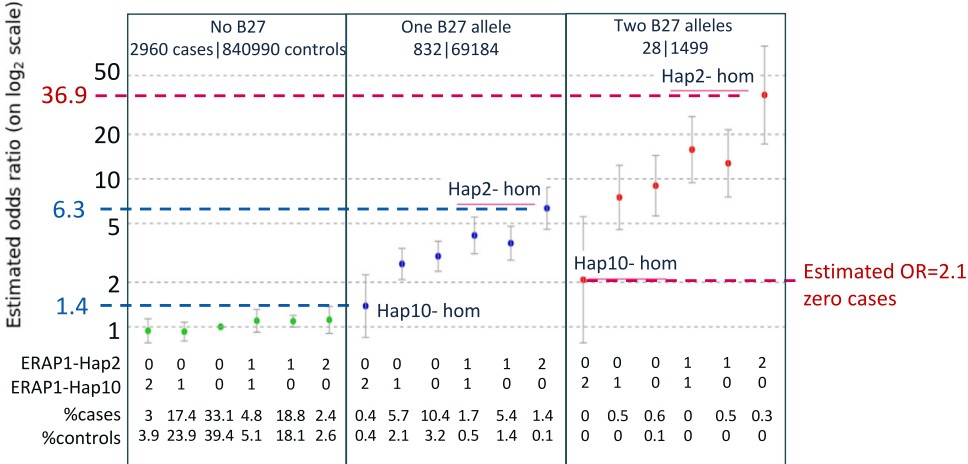

**Fig. 4 | The combined risk for AU with *HLA-B\*27* and *ERAP1*-haplotyes.** The effect of homozygous and heterozygous *ERAP1* haplotypes Hap2 and Hap10 on different *HLA-B\*27* backgrounds. Zero *HLA-B* and *ERAP1* protection haplotypes combination (two copies of Hap10 and no copies of Hap2) was defined as the reference risk genotype (i.e. OR = 1, first column on left panel). The assessed risk of the *ERAP1* Hap10 and Hap2 combinations on a B\*27 negative background is shown (left panel and supplementary Data 5). Middle panel is the same as left panel, but for one copy of *HLA-B\*27*. Right panel is the same as left panel, but for two copies of *HLA-B\*27*.

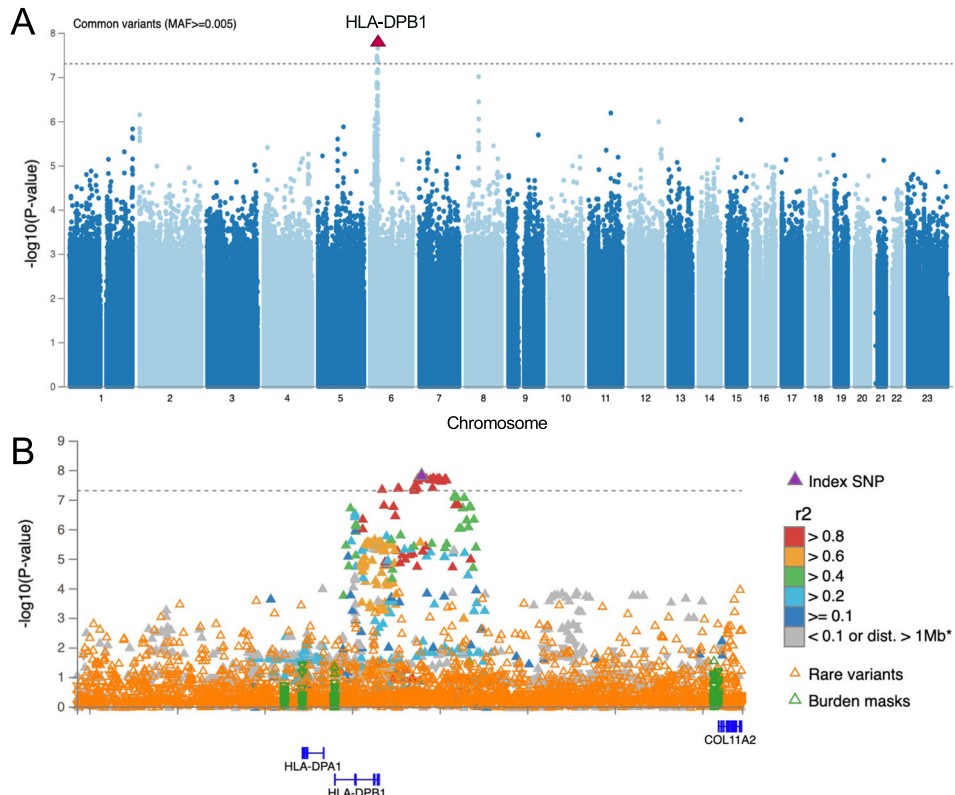

**Fig. 5 | HLA-*DPB1* is a significant risk for B\*27-neg AU. A** Manhattan plot depicting the -log10(*P* value) for all common variants (*y*-axis) across all chromosomes (*x*-axis). *HLA-DPB1* top risk signal is shown by an upward red triangle on chromosome five. **B** A locus zoom plot showing all common and rare signals on *HLA-DPB1*. Genome-wide significant threshold of 5e−08 is represented by a dashed gray line, above which there is a stretch of high LD variants downstream to *HLA-DPB1*. Association models were run with age, age2, sex and age × sex, and 10 ancestry-informative principal components as covariates. *P* values are uncorrected and are from two-sided tests performed using approximate Firth logistic regression.

spondyloarthropathies including ankylosing spondylitis, psoriatic arthritis, arthritis associated with IBD and reactive arthritis[29–31]. Until now, it has been difficult to disentangle AU analysis from the other diseases that are well recorded and that allow much larger studies. With EHR data for almost one million samples, we were able to study the genetics of AU specifically by focusing on individual ICD diagnosis codes and removing the most common co-morbidities from the cohort for better elucidation of the genetic signals. We include in our analyses whole-exome sequence data for the full set of samples, allowing discovery of genes in which rare coding changes impact AU risk. We incorporate all ancestries into comprehensive analyses that dissect the contributions from different ancestral population.

**Table 4 | IPMK and IDO2 genes are associated with B*27-neg AU**

| Gene Burden Mask | Cohort | ALL ancestries | | | | EUR only | | | | AFR only | | | |
|---|---|---|---|---|---|---|---|---|---|---|---|---|---|
| | | OR (LCI-UCI) | Pval | Cases ref\|het\|alt | Controls ref\|het\|alt | OR (LCI-UCI) | Pval | Cases ref\|het\|alt | Controls ref\|het\|alt | OR (LCI-UCI) | Pval | Cases ref\|het\|alt | Controls ref\|het\|alt |
| IPMK pLoF and damaging missense (5/5) AF <0.1% | Full cohort | 9.4 (4.5-19.9) | 4.4E-09 | 3839\|11\|0 | 916138\|411\|0 | 8.04 (2.9-22.6) | 7.7E-05 | 3174\|6\|0 | 826121\|336\|0 | 21.14 (3.8-116.0) | 4.64E-04 | 417\|3\|0 | 42243\|27\|0 |
| | B*27-neg cohort | 10.1 (4.8-22.2) | 2.3E-09 | 2973\|11\|0 | 844328\|381\|0 | 11.32 (3.8-33.6) | 1.25E-05 | 2357\|6\|0 | 756860\|308\|0 | 25.49 (4.6-142.4) | 2.24E-04 | 400\|3\|0 | 41409\|27\|0 |
| | B*27-pos cohort | 0.34 (0.01-11.4) | 5.47E-01 | 787\|0\|0 | 63821\|30\|0 | 0.36 (0.01-12.0) | 5.57E-01 | 752\|0\|0 | 62190\|28\|0 | NA | | | |
| IDO2 pLoF only AF <0.1% | Full cohort | 3.61 (2.3-5.8) | 6.16E-08 | 3822\|27\|1 | 915111\|1433\|5 | 2.96 (1.2-7.6) | 2.30E-02 | 3174\|6\|0 | 825771\|686\|0 | 2.15 (1.3-3.5) | 1.93E-03 | 400\|19\|1 | 41598\|667\|5 |
| | B*27-neg cohort | 3.28 (2.0-5.3) | 9.5E-07 | 2958\|25\|1 | 843322\|1382\|5 | 2.32 (0.8-6.5) | 1.10E-01 | 2359\|4\|0 | 756525\|643\|0 | 2.32 (1.4-3.8) | 9.12E-04 | 383\|19\|1 | 40771\|660\|5 |
| | B*27-pos cohort | 9.56 (1.4-67.0) | 2.31E-02 | 824\|2\|0 | 67243\|48\|0 | 7.21 (0.9-59.5) | 6.64E-02 | 776\|2\|0 | 64911\|43\|0 | NA | | | |

Association models were run with age, age², sex and age × sex, and 10 ancestry-informative principal components as covariates. P values are uncorrected and are from two-sided tests performed using approximate Firth logistic regression. Numbers below the cases and controls columns represent counts of individuals with homozygote reference, heterozygote and homozygous alternative genotypes, respectively.

Consequently, we explore the underlying genetics of *HLA-B*27* uveitis and distinguish *HLA-B*27* positive and *HLA-B*27* negative uveitis as two genetically distinct diseases. Although clinical manifestations overlap considerably, B*27-pos AU is typically characterized by more robust inflammation and is more likely to recur than B*27-neg AU[32,33].

When stratifying the cohorts by *HLA-B*27*, we observe a limited range of B*27 carriers around 25%, much lower than the 50% previously reported[9]. Previous reports suggested around 50% B*27-carriers but were based on smaller studies[34]. Also, the AFR population in our datasets could also have reduced the proportion of *HLA-B*27* carriers having much smaller occurrence of B*27 in the AFR ancestry (-1.7% of AFR controls). The large cohorts at hand allowed us to still study this much smaller cohort of B*27-pos and elucidate the clear effect of *ERAP1* as the strongest factor affecting AU risk and protection in *HLA-B*27* AU. *ERAP1* has been previously reported as nominally conferring risk for AU in AS cases[12,14]. However, previous studies were underpowered to detect a significant the AU-*ERAP1* signal, with cases having a major AS diagnosis, which made it hard to disentangle AS diagnosis from AU analysis.

Different combinations of non-synonymous SNPs give rise to the ten main *ERAP1* haplotypes with differences in enzymatic activity and/or expression levels[17]. By utilizing the full size of the cohort and dissecting the samples into the ten common *ERAP1* haplotypes, we were able to see the strong protection of two copies of Hap10, that may offer disease prevention even for individuals carrying two copies of the strongest B*27 AU risk alleles.

The *ERAP1* low-activity low-expression Hap10 haplotype was previously shown to protect against AS, and to express a different B*27 peptidome than the risk Hap2 by providing reduced trimming of peptides[17,35]. In this context, the strong Hap10 homozygous protection that we observe for AU suggests that *ERAP1* expression and enzymatic activity are lowest in these carriers. This also suggests that the peptidome shaped by Hap10 could be deficient in the antigenic peptide(s) that active the immune response in AU cases.

The stratification by *HLA-B*27* also enabled us to observe a clear common genetic risk for B*27-neg uveitis in the form of class-II *HLA-DPB1*. This signal was distinct from B*27-pos AU and points to a distinct mechanism for the two diseases. While B*27-pos AU is driven by antigenic-peptide(s), where we hypothesize the mechanism of *ERAP1* variants is affecting the peptides available for presentation in *HLA-B*27*, the mechanism of disease in B*27-neg AU may differ. The participation of a class-II gene, either directly by a specific allele, or indirectly through a change in expression, suggests a mechanism similar to celiac disease, where an exogenous immunogenic factor initiates the cascade that leads to pathogenicity[36]. In the case of AU, this might be something like a cataract surgery that exposes immune cells to tissues that are normally sequestered (the crystalline lens), however, further investigations are required to confirm such a hypothesis.

The availability of rare variants from exome sequencing, in addition to genotyping and imputation, allowed us to identify two genes where rare genetic variants affect AU risk. In the case of *IPMK* (Inositol Polyphosphate Multikinase), we find that either missense or loss-of-function variants combine together to show increased risk of disease. *IPMK*'s catalytic activities yields water-soluble inositol polyphosphates and is considered a signaling hub in mammalian cells that coordinates the activity of various signaling networks including regulating the TLR-induced innate immunity[37]. *IPMK* promotes Toll-like receptor–induced inflammation by stabilizing *TRAF6*, the Tumor Necrosis Factor Receptor–Associated Factor 6, that is a critical mediator of TLR signaling[38]. While this might be a valid mechanism affecting AU pathology, it remains to be seen exactly how *IPMK* affects AU risk.

For *IDO2* (Indoleamine 2,3-dioxygenase 2) we observe significant risk through a clear loss-of-function mechanism that is shared between the studied cohorts. *IDO2*, like *IDO1*, was reported necessary for the differentiation of regulatory T cells in vitro and has been shown to play

a pro-inflammatory role in the development of B cell-mediated auto-immune arthritis[39,40]. It is then likely that the loss of *IDO2* might disrupt T-cell regulation and affect the T-Cell mediated response in the anterior chamber, contributing to the patho-mechanism of AU. While a precise role for *IPMK* and *IDO2* in regulating immune tolerance in the anterior segment remains opaque, it is relevant to note that both proteins are expressed locally. Single-nucleus RNA sequencing data made available through The Broad Institute demonstrate *IPMK* and IDO2 expression in both the iris (irido-) and ciliary body (-cyclitis) (figures S8-S11, https://singlecell.broadinstitute.org/single_cell/study/SCP1841/).

Taken together, these results highlight the underlying and distinct genetics of B\*27-pos and B\*27-neg AU, presenting them as two genetically distinct diseases. We further identify the protection of *ERAP1*-Hap10, which raises the enticing prospect of *ERAP1*'s therapeutic potential in the management of AU. This is particularly relevant in B\*27-pos AU where recurrent episodes of inflammation, difficult to control with topical steroids, put patients at increased risk of vision-threatening complications. Last, we uncover several risk genes for B\*27-neg AU: including a common locus that affects AU risk in *HLA-DPB1*, as well as two risk genes and several candidate genes affecting disease risk through rare variation causing loss-of-function (as in *IDO2*) and/or changes to the protein sequence (as in *IPMK*), thus further elucidating the genetic risks for AU.

## Methods

### Study populations

Genome-wide association analyses were performed in eight cohorts including the U.K. Biobank cohort[41] and the Geisinger Health System MyCode cohort[42]. Others datasets include: 29,237 from the Malmö Diet and Cancer Study[43],

41,537 participants from the University of Pennsylvania Penn Medicine BioBank[44],

29,845 participants from the Mount Sinai BioMe BioBank[45],

49,071 from the Colorado Center for Personalized Medicine Biobank[46], and 40,217 from the UCLA ATLAS Community Health Initiative[47,48]. We also included 115,418 participants from the MAYO-RGC Project Generation, which brings together the Mayo Clinic Biobank (*N* = 53,227)[49] as well as 30 Mayo-based disease registries/studies who were successfully sequenced. This study was reviewed and approved by the Mayo Clinic IRB (#09-007763).

We included 829,865 participants of European ancestry, 42,790 of African ancestry, 13,870 of South Asian ancestry, 9305 of East Asian ancestry, 18,868 with ancestry from the Americas, and 5653 of other ancestries, for whom genotyping, exome-sequencing data and phenotype data were available (full breakdown of cohorts in supplementary Data 9). Cases were selected based on the "ICD10: H20 Iridocyclitis" diagnosis code, controls were defined as individuals without the ICD10: H20 code.

### Ethical compliance

Ethical approval for the UK Biobank was previously obtained from the North West Center for Research Ethics Committee (11/ NW/ 0382). The work described herein was approved by UK Biobank under application number 26041. Approval for Geisinger Health System MyCode analyses was provided by the Geisinger Health System Institutional Review Board under project number 2006-0258. Informed consent was obtained for all study participants. Appropriate consent for the University of Pennsylvania Penn Medicine BioBank was obtained from each participant regarding storage of biological specimens, genetic sequencing and genotyping, and access to all available EHR data. This study was approved by the Institutional Review Board of the University of Pennsylvania and complied with the principles set out in the Declaration of Helsinki. All subjects participating in the MAYO-RGC Project Generation

provided informed consent for use of specimens and data in genetic and health research and ethical approval for Project Generation was provided by the Mayo Clinic IRB (#09-007763). Ethical approval and consent for the Colorado Center for Personalized Medicine Biobank was reviewed and approved by the Colorado Multiple Institutional Review Board (#15-0461). All research performed in the UCLA ATLAS Community Health Initiative study conformed with the principles of the Helsinki Declaration. All individuals provided written informed consent to the original recruitment of the UCLA ATLAS Community Health Initiative. Patient Recruitment, Sample Collection for Precision Health Activities at UCLA is an approved study by the UCLA Institutional Review Board (UCLA IRB). IRB#17-001013. All research performed in this study uses de-identified data (without any Protected Health Information data) with no possibility of re-identifying any of the participants. The Mount Sinai BioMe BioBank study protocols were approved by the institutional review board of the Icahn School of Medicine at Mount Sinai. Written informed consent was obtained for all study participants. All participants in the Malmö Diet and Cancer Study were provided written informed consent and the study was approved by the Lund University Ethics Committee (MDC LU 51-90) and for the cadmium substudy (2009/633).

### Exome Sequencing and whole-genome genotyping

For analyses of common variants, we used array genotyping data and imputation performed with the use of the TOPMed reference panel[50,51]. Exome sequencing was performed at the Regeneron Genetics Center using a custom automated sample preparation approach. Samples were captured with IDT xGen v1 or Twist Comprehensive Exome probes and sequenced using Illumina HiSeq 2500-v4 or Illumina NovaSeq instruments, with 75-bp paired-end reads and two index reads. The GRCh38 human genome reference sequence and Ensembl, version 85, gene definitions were used for variant identification and annotation. For the COLORADO, MAYO-CLINIC and UCLA cohorts sequenced with Twist, probes also included the Twist Diversity SNP panel, for which multi-point refinement was conducted using GLIMPSE prior to further genotype QC and imputation[52]. For exome coding variants, we classified variants from most to least deleterious in the following order: frameshift, stop–gain, stop–loss, splice acceptor, splice donor, in-frame insertion or deletion (indel), missense, and other annotations. Frameshift, stop–gain, stop–loss, splice-acceptor, and splice-donor alleles were categorized as predicted loss-of-function variants. We classified missense variants using computer modeling to predict functional effects with five algorithms: SIFT[53], Polyphen-2 HDIV[54], Polyphen-2 HVAR[54], LRT[55] and MutationTaster[56]. To account for the fact that different genes have different types and frequencies of potentially causative variants, we used the functional annotation of the variants in each gene to generate seven pseudo-genotypes based on the combined variant burden: predicted loss-of-function variants; predicted loss-of-function variants plus missense variants that were predicted to be deleterious by five of five algorithms; predicted loss-of-function variants plus missense variants that were predicted to be deleterious by at least one of five algorithms; predicted loss-of-function variants plus any missense variants; missense variants that were predicted to be deleterious by five of five algorithms; missense variants that were predicted to be deleterious by at least one of five algorithms; and finally, any missense variants at all (these categories are similar to those used previously)[57]. We used the alternative allele frequency and functional annotation of each variant to generate seven genotypes based on the combined variant burden: pLoF variants with an alternative-allele frequency thresholds of 1%, 0.1%, 0.01% and singletons, pLoF variants plus missense variants that were predicted to be deleterious and had an alternative-allele frequency thresholds of 1%, 0.1%, 0.01% and singletons.

## Statistical analysis

We estimated associations between genotypes and phenotypes by fitting linear regression models (for quantitative traits) or Firth bias-corrected logistic regression models (for binary traits) using the REGENIE software, version 2 + [58]. Analyses were stratified according to cohort and ancestry and were adjusted for age, age squared, sex, age-by-sex, and age squared–by–sex interaction terms; experimental batch-related covariates; the first 10 common variant–derived genetic principal components; the first 20 rare variant–derived principal components; and a polygenic score generated by REGENIE, which robustly adjusts for relatedness and population structure[58]. We performed a meta-analysis of association results across cohorts and ancestries with a fixed-effect inverse-variance–weighted approach. We report results for TOPMED imputed data for common variants defined by minor allele frequency greater than 0.5%, and we report results for exome sequenced rare coding variants that had a minor allele count greater than five in both cases and controls. For gene burden analyses, we tested each of the variant-burden categories mentioned above at four thresholds of alternate-allele frequencies: alternative-allele frequencies of less than 1%; alternative-allele frequencies of less than 0.5%; alternative-allele frequencies of less than 0.1%; and alternative-allele frequencies of less than 0.01%. These seven categories and four thresholds produce 28 pseudo-genotypes for each gene, but they are not fully independent of one another, given the overlapping annotations and frequency thresholds. Thus, we calculated an appropriate adjusted Bonferroni significance level for these variant-burden tests, using a method recommended by a review of multiple-testing correction methods in non-independent genetic tests[18,59]. Calculating the effective number of independent tests based on the correlation matrix of these variant-burden tests in our meta-analysis resulted in a value of 9.002158 tests per gene, which, when multiplied by the number of genes tested (19,446) and used as a correction factor for an alpha level of 0.05, resulted in an exome-wide level of significance at a $P$ value of 2.86e−07.

### Reporting summary

Further information on research design is available in the Nature Portfolio Reporting Summary linked to this article.

## Data availability

The primary analysis summary data generated in this study have been deposited in the GWAS Catalog database under accession code [GCST ID: GCST90295958, GCST90295959]. Individual-level sequence data have been deposited with UK Biobank and will be freely available to approved researchers, as done with other genetic datasets to date. Individual-level phenotype data are already available to approved researchers for the surveys and health-record datasets from which all our traits are derived. Instructions for access to UK Biobank data is available at https://www.ukbiobank.ac.uk/enable-your-research. Summary statistics from UK Biobank trait are available in the GWAS Catalog (accession IDs are listed in the tables description sheet available in the supplementary data tables excel file). Exome sequencing and genotyping data used for meta-analysis from additional cohorts such as the Geisinger Health System MyCode, the Malmö Diet and Cancer Study, the University of Pennsylvania Penn Medicine BioBank, the Mount Sinai BioMe BioBank and the Colorado Center for Personalized Medicine Biobank, can be made available to qualified, academic, non-commercial researchers upon request via a Data Transfer Agreement with the respective research institute. Aggregate data from the UCLA ATLAS Community Health Initiative can be made available to qualified, academic, non-commercial researchers on a collaborative basis upon request. As described in Backman et al.9, the HapMap3 reference panel was downloaded from https://ftp.ncbi.nlm.nih.gov/hapmap/, GnomAD v3.1 VCFs were obtained from https://gnomad.broadinstitute.org/downloads, and VCFs for TOPMED Freeze 8 were obtained from

dbGaP as described in https://topmed.nhlbi.nih.gov/topmed-whole-genome-sequencing-methods-freeze-8.

## Code availability

The REGENIE software for whole genome regression, which was used to perform all genetic association analysis, is available at https://github.com/rgcgithub/regenie. GCTA v1.91.7 was used for approximate conditional analysis. SHAPEIT4.2.0 was used for phasing of SNP array data. Imputation was completed with IMPUTE5. We use Plink1.9/2.0 for genotypic analysis. FINEMAP 1.4.1 and SuSiE 0.12.27 were used for fine-mapping, and genetic correlations were calculated using LDSC version 1.0.1 with annotation input version 2.2. R Statistical Computing 4.1 was used including packages with visualization tools, statistical and data processing libraries (e.g. base R 4.1, dplyr 1.1.2, ggplot2 3.3.6, data.table 1.14.2).

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

## Acknowledgements

We thank the UK Biobank team, their funders, the dedicated professionals from the member institutions who contributed to and supported this work, and the UK Biobank participants. The exome sequencing was funded by the UK Biobank Exome Sequencing Consortium (i.e., Bristol Myers Squibb, Regeneron, Biogen, Takeda, AbbVie, Alnylam, AstraZeneca and Pfizer). This research has been conducted using the UK Biobank Resource under application number 2604. We thank the MyCode Community Health Initiative participants for taking part in the GHS DiscovEHR collaboration. We gratefully acknowledge the support of the Institute for Precision Health, participating patients from the UCLA ATLAS Precision Health Biobank, UCLA David Geffen School of Medicine, UCLA Clinical and Translational Science Institute grant number UL1TR001881, and UCLA Health. We acknowledge the Penn Medicine BioBank (PMBB) for providing data and thank the patient-participants of Penn Medicine who consented to participate in this research program. We would also like to thank the Penn Medicine BioBank team and Regeneron

Genetics Center for providing genetic variant data for analysis. The PMBB is approved under IRB protocol# 813913 and supported by Perelman School of Medicine at University of Pennsylvania, a gift from the Smilow family, and the National Center for Advancing Translational Sciences of the National Institutes of Health under CTSA award number UL1TR001878. Furthermore, the authors would like to acknowledge the support from Lund University Infrastructure grant"Malmö population-based cohorts" (STYR 2019/2046). We also would like to acknowledge the volunteer participants from the Mt. Sinai School of Medicine BioME study, the Mayo Clinic Project Generation, and the Colorado Center for Precision Medicine. This research received funding from Regeneron Pharmaceuticals, Inc.

## Author contributions

S.G., A.M., S.M.H., R.W., V.R., N.P., V.K.P., E.C., M.L., R.H., J.F., B.C., A.J.L., O.M., C.G.M., T.V., J.W., C.R., B.S., A.B., E.A.S. and G.C. reviewed the manuscript for important intellectual content and approved the manuscript submitted for publication. Conceptualization: S.G. and G.C. Genetic analysis: S.G., A.M., S.M.H., R.W., V.R., N.P., V.K.P., Phenotype preparation and harmonization: S.G., A.M., R.W. Analytical pipeline development: S.G., A.M., R.W., V.R., N.P., V.K.P., E.A.S. Data curation: S.G., A.M., S.M.H., R.W., V.K.P., E.A.S., G.C. Funding acquisition: A.B., G.C. Project administration: M.L., E.C., R.H. Supervision: A.B., G.C., E.A.S. Writing—original draft: S.G., S.M.H., A.M., E.A.S., C.G.M., G.C. All authors contributed to study design and oversight and reviewed the final version of the manuscript.

## Competing interests

S.G., A.M., S.M.H., R.W., V.R., N.P., V.K.P., E.C., M.L., R.H., J.F., B.C., A.J.L., C.G.M., T.V., J.W., C.R., B.S., A.B., E.A.S. and G.C. are current employees and/or stockholders of Regeneron Genetics Center or Regeneron Pharmaceuticals. All other authors declare no conflict of interest.

## Additional information

## Regeneron Genetics Center

Aaron Zhang[1], Adam J. Mansfield[1], Adam Locke[1], Aditeya Pandey[1], Adrian Campos[1], Arkopravo Ghosh[1], Alexander Gorovits[1], Alexander Lopez[1], Alicia Hawes[1], Alison Fenney[1], Amelia Averitt[1], Amit Joshi[1], Amy Damask[1], Andrew Bunyea[1], Andrey Ziyatdinov[1], Anita Pandit[1], Ann Perez-Beals[1], Anna Alkelai[1], Anthony Marcketta[1], Antoine Baldassari[1], Arden Moscati[1], Ariane Ayer[1], Arthur Gilly[1], Ayesha Rasool[1], Aysegul Guvenek[1], Benjamin Geraghty[1], Benjamin Sultan[1], William Palmer[1], Bin Ye[1], Blair Zhang[1], Boris Boutkov[1], Brian Hobbs[1], Caitlin Forsythe[1], Carlo Sidore[1], Charles Paulding[1], Chenggu Wang[1], Christina Beechert[1], Christopher Gillies[1], Chuanyi Zhang[1], Cristen J. Willer[1], Dadong Li[1], Deepika Sharma[1], Eli Stahl[1], Eliot Austin[1], Eric Jorgenson[1], Erin D. Brian[1], Ernst Mayerhofer[1], Esteban Chen[1], Evan Edelstein[1], Evan K. Maxwell[1], Gannie Tzoneva[1], George Hindy[1], George Mitra[1], Gina Solari[1], Gisu Eom[1], Hang Du[1], Hossein Khiabanian[1], Jack Kosmicki[1], Jacqueline Otto[1], Jaimee Hernandez[1], Janice Clauer[1], Jason Mighty[1], Jeffrey C. Staples[1], Jennifer Rico-Varela[1], Jessie Brown[1], Jing He[1], Jingning Zhang[1], Joana Revez[1], Jody Hankins[1], Joelle Mbatchou[1], Johannie Rivera-Picart[1], John Silver[1], Jonas Bovijn[1], Jonathan Marchini[1], Jonathan Ross[1], Jose Bras[1], Joseph Herman[1], Joshua Backman[1], Ju Guan[1], Juan Rodriguez-Flores[1], Justin Mower[1], Karl Landheer[1], Kathie Sun[1], Kathy Burch[1], Kayode Sosina[1], Kia Manoochehri[1], Kimberly Skead[1], Krishna Pawan Punuru[1], Kristy Guevara[1], Kuan-Han Wu[1], Kyoko Watanabe[1], Lance Zhang[1], Laura M. Cremona[1], Lauren Gurski[1], Lei Chen[1], Liron Ganel[1], Luanluan Sun[1], Lukas Habegger[1], Manasi Pradhan[1], Manav Kapoor[1], Manuel Allen Revez Ferreira[1], Marcus B. Jones[1], Maria Sotiropoulos Padilla[1], Maria Cristina Suciu[1], Maya Ghoussaini[1], Mary Haas[1], Michael Lattari[1], Michael Kessler[1], Michelle G. LeBlanc[1], Michelle Pagan[1], Mira Tang[1], Moeen Riaz[1], Mona Nafde[1], Mudasar Sarwar[1], Nadia Rana[1], Nan Lin[1], Neelroop Parikshak [1], Niek Verweij[1], Nilanjana Banerjee[1], Nirupama Nishtala[1], Olga Krasheninina[1], Oliver Delaneau[1], Olukayode Sosina[1], Parsa Akbari[1], Peter Dornbos[1], Peter VandeHaar[1], Prathyusha Challa[1], Priyanka Nakka[1], Randi Schwartz[1], Raymond Reynoso[1], Razvan Panea[1], Ricardo Schiavo[1], Rita Guerreiro[1], Rouel Lanche[1], Rujin Wang[1], Sahar Gelfman[1], Sailaja Vedantam[1], Salvador Romero Martinez[1], Sam Choi[1], Samantha Zarate[1],

**Sameer Malhotra**[1], **Samuel Hart**[1], **Sanjay Sreeram**[1], **Sarah E. Wolf**[1], **Sarah Graham**[1], **Scott Vrieze**[1], **Sean O'Keeffe**[1], **Sean Yu**[1], **Sheila Gaynor**[1], **Silvia Alvarez**[1], **Suganthi Balasubramanian**[1], **Sujit Gokhale**[1], **Sunilbe Siceron**[1], **Suying Bao**[1], **Tanima De**[1], **Timothy Thornton**[1], **Tommy Polanco**[1], **Tyler Joseph**[1], **Valentina Zavala**[1], **Veera Rajagopal**[1], **Vijay Kumar**[1], **Vrushali Mahajan**[1], **William J. Salerno**[1], **Xiaodong Bai**[1], **Yuxin Zou**[1], **Zhenhua Gu**[1], **Adolfo Ferrando**[1], **Alan Shuldiner**[1], **Andrew Deubler**[1], **Aris Economides**[1], **Giovanni Coppola** ®[1,4] ✉, **Gonçalo Rocha Abecasis**[1], **Jeffrey G. Reid**[1], **John D. Overton**[1], **Katherine Siminovitch**[1], **Luca A. Lotta**[1], **Lyndon J. Mitnaul**[1], **Michael Cantor**[1] & **Aris Baras** ®[1]

