## [Peer Review File · Nature Communications]

A Large Meta-Analysis Identifies Genes Associated with Anterior UveitisREVIEWER COMMENTS

Reviewer #1 (Remarks to the Author):

Gelfman et al has a fascinating article of the largest meta-analysis to date, focusing on anterior uveitis, with 900,000 controls and 3,850 cases of diverse ancestries. Two subtypes are delineated, and found a HLA-B*27 dependent protective signal in ERAP1. No rare variants were identified, and the same results were found in restriction analysis to those of EU descent, as well as the large meta-analysis of 6/8 cohorts. Burden testing including loss of function and missense identified IMPK and IDO2. Singleton analysis identified variants in the genes ACHE , STXBP2 , ADGRF5. The article is well written and clearly done, and will be of use to the field considering the diverse ancestries and large scale study. A few minor questions:

1. When the singleton analysis was performed, variants were found in three genes. ACHE , STXBP2 , ADGRF5. Did ACHE and ADGRF5 show a specific ancestry like STXBP2 did (where all 9 cases were of AFR descent)?
2. You list that there is a different and higher than normal percentage of the subtype of B*27 (carrying the B*27 allele, or the B*27-pos cohort) (22.2%)- can you speculate or have any idea as to why it is much higher in your dataset than previously reported? Has this finding been found elsewhere?
3. Which tissues were examined in GTEX specifically for eQTL analysis- line 184, 156?
4. In the "related data" file, in the last line of the IPMK paragraph it looks like something is completely missing? It cuts out in the middle of the sentence: "A Fisher's exact test was..."
5. 6/9 carriers of some of the rare burden gene analysis on STXBP2 originate from the UPENN cohort – any ideas why?
6. Can you include the data of the general AF of rs151088117 in the AFR population because the 24/28 carriers are all from AFR descent? You mention that there is a higher AF, is it significantly higher? Where did you check AF?

Reviewer #2 (Remarks to the Author):

The manuscript titled: "A Large Meta-Analysis Identifies Genes Associated with Anterior Uveitis" analyzed exome sequencing data and performed a meta-analysis of eight independent cohorts consisting of 3,850 AU cases and 916,549 24 controls.

The overall strengths of this paper include: it is the genetic largest study of AU, the paper is very well written with a good rationale, there is a clear presentation of results, as well as the identification of new common genetic signals and genes harboring rare variants linked to AU. The distinction of HLA-B*27 positive and negative AU was interesting.

Specific Comments

- For the rare variant analysis, please provide more information in the relevant supplementary tables on each of the variants detected within IPMK, IDO2 as well as the other genes ACHE, STXBP2 and ADGRF5. This information includes the coding change (eg c. Position C>G) and relevant transcript and protein change (eg p.Ser255Gly), and whether the variant is present in a specific protein domain. Please include the frequency of the variants in the population database Gnomad/Exac, or whether the variants are listed as pathogenic in Clinvar and also include analysis using bioinformatic prediction of variant consequence (through tools such as CADD, REVEL, polyphen etc) as this will support whether the variants are rare in the population and whether variants, in particular the missense variants, have an impact on protein function.
- Would it be possible to provide some functional data to support the role of IPMK, IDO2, ACHE, STXBP2 or ADGRF5 in Anterior Uveitis?

- Some further discussion on the implications of the distinction between HLA-B*27 positive and negative AU would be useful, including how these findings may be translated in the future.
- While this is a well written paper, it would may useful to further highlight the key strengths of the work in context of the broader readership of the journal.

Dear reviewers:

We would like to thank the reviewers for the comments and the opportunity to improve our manuscript through the responses. We were pleased that the reviewers felt the manuscript is comprehensive, well-written and adds to the understanding of Anterior Uveitis. We have now addressed the important concerns raised by both reviewers and believe that the added results strengthen this work. Based on the remarks from the reviewers, we performed several additional analyses, added an additional supplementary table (S9) and figure (S12) and revised the Results, Discussion, and Extended Data sections.

Below, we address each of the specific reviewer's comments and cite the location of the edits corresponding to the marked-up version of our resubmission.

Reviewer #1 (Remarks to the Author):

Gelfman et al has a fascinating article of the largest meta-analysis to date, focusing on anterior uveitis, with 900,000 controls and 3,850 cases of diverse ancestries. Two subtypes are delineated, and found a HLA-B*27 dependent protective signal in ERAP1. No rare variants were identified, and the same results were found in restriction analysis to those of EU descent, as well as the large meta-analysis of 6/8 cohorts. Burden testing including loss of function and missense identified IMPK and IDO2. Singleton analysis identified variants in the genes ACHE , STXBP2 , ADGRF5. The article is well written and clearly done, and will be of use to the field considering the diverse ancestries and large scale study. A few minor questions:

1. When the singleton analysis was performed, variants were found in three genes. ACHE , STXBP2 , ADGRF5. Did ACHE and ADGRF5 show a specific ancestry like STXBP2 did (where all 9 cases were of AFR descent)?

- This is a very good question by the reviewer that we have now clarified in the manuscript as well. *ACHE* singleton carriers are of EUR-ancestry only. For *ADGRF5*, it replicates nominal risk in both EUR and AFR ancestries, albeit the number of AFR carriers is very small (only one case). We have added this information to the updated manuscript (supplementary figure S12 and Extended data document, section "Additional testing of candidate genes", p.1-2; lines 24-42)

2. You list that there is a different and higher than normal percentage of the subtype of B*27 (carrying the B*27 allele, or the B*27-pos cohort) (22.2%)- can you speculate or have any idea as to why it is much higher in your dataset than previously reported? Has this finding been found elsewhere?

- The reviewer is referring to the proportion of B*27 carriers in our data, as reported in the Results section lines 145-148: "B*27-pos cohort consisted of 856 AU cases and 70,107 controls, suggesting that 22.2% of the analyzed AU cases carry the B*27 allele. This is a significant enrichment compared to the 7.7% B*27 carriers in the controls, similar to the 6%-8% expected HLA-B*27 frequency in general population in the US²⁰." In the large HER cohorts we describe in this study, we actually find lower, and not higher, proportion of B*27 carriers (22.2%) compared published reports of ~50%. We speculate that: 1) previous reports were based on smaller studies and, 2) the AFR population in our datasets has a very low proportion of B*27 carriers in general: 741/42,273 or ~1.8% out of the overall AFR controls in our datasets. This low

proportion reduces the proportion of HLA-B*27 carriers in the full, ALL ancestries cohort. We have incorporated the above information to the Discussion section (p.11; lines 314-316). Last, Table S8 in the updated manuscript details a breakdown of the number of cases and controls per cohort, ancestry and proportion of B*27 carriers, which presents the lower proportion of B*27 carriers in the AFR ancestry.

3. Which tissues were examined in GTEx specifically for eQTL analysis- line 184, 186?

- rs30187 is a strong eQTL signal downregulating *ERAP1* in all 44 tissues represented in GTEx, which unfortunately does not include eye tissues but includes whole blood 4.4e-78, skeletal muscle 4.6e-49, lung 6.1e-46, etc. rs10050860 is not as strong an eQTL as rs30187, but still shows strong *ERAP1* downregulation effects in 26/28 tissues (including skeletal muscle 5.0e-43, whole blood 9.8e-31 and esophagus 1.0e-14), and weak increased expression in the tibial artery ($p=1.4e-4$) and ovary ($p=5.2e-6$). This information has been added to the manuscript (p.7; lines 186-189).

4. In the “related data” file, in the last line of the IPMK paragraph it looks like something is completely missing? It cuts out in the middle of the sentence: “A Fisher’s exact test was...”

- We thank the reviewer for catching this mistake, we have corrected the file accordingly.

5. 6/9 carriers of some of the rare burden gene analysis on STXP2 originate from the UPENN cohort – any ideas why?

- This is a good question by the reviewer that we have debated as well. Since the UPENN cohort has a large proportion of AFR ancestry compared with most other cohorts except the SINAI cohort (supplementary table S8), there’s a high chance for AFR carriers. We also considered that relatedness between those carriers will affect this result, but based on a genetic analysis we found they are unrelated.

6. Can you include the data of the general AF of rs151088117 in the AFR population because the 24/28 carriers are all from AFR descent? You mention that there is a higher AF, is it significantly higher? Where did you check AF?

- This is indeed valuable information to add to the manuscript, since the population differences are strong. The population specific allele frequency information for rs151088117 is based on the gnomAD population database V2 including 141,456 samples (https://gnomad.broadinstitute.org/variant/8-39847312-A-T?dataset=gnomad_r2_1). The gnomAD database presents AFR AF=0.0107 based on 22,440 AFR alleles, and non-finish EUR AF=0.0 considering 121,344 alleles, depicting a very distinct AFR signal. We have incorporated this information into the current manuscript (Extended data document, p.2; lines 55-60 and supplementary table S9).

Reviewer #2 (Remarks to the Author):

The manuscript titled: "A Large Meta-Analysis Identifies Genes Associated with Anterior Uveitis" analyzed exome sequencing data and performed a meta-analysis of eight independent cohorts consisting of 3,850 AU cases and 916,549 24 controls.

The overall strengths of this paper include: it is the genetic largest study of AU, the paper is very well written with a good rationale, there is a clear presentation of results, as well as the identification of new common genetic signals and genes harboring rare variants linked to AU. The distinction of HLA-B*27 positive and negative AU was interesting.

Specific Comments

- For the rare variant analysis, please provide more information in the relevant supplementary tables on each of the variants detected within *IPMK*, *IDO2* as well as the other genes *ACHE*, *STXBP2* and *ADGRF5*. This information includes the coding change (eg c. Position C>G) and relevant transcript and protein change (eg p.Ser255Gly), and whether the variant is present in a specific protein domain. Please include the frequency of the variants in the population database Gnomad/Exac, or whether the variants are listed as pathogenic in Clinvar and also include analysis using bioinformatic prediction of variant consequence (through tools such as CADD, REVEL, polyphen etc) as this will support whether the variants are rare in the population and whether variants, in particular the missense variants, have an impact on protein function.

- We agree with the reviewer that this is important information that will help to follow-up on the genetic results with well-constructed functional studies. We therefore constructed a supplementary table that has the full breakdown of all case variants in each of the top gene burdens, and with all relevant annotations for each variant. This information is now incorporated to the updated manuscript as supplementary table S9.

- Would it be possible to provide some functional data to support the role of *IPMK*, *IDO2*, *ACHE*, *STXBP2* or *ADGRF5* in Anterior Uveitis?

- This question of roles of the identified genes is certainly of great interest and we appreciate the reviewer's inquiry on the subject. In the manuscript, we include a paragraph speculating on the functional roles for *IPMK* and *IDO2* in immune response in the Discussion section (p.13; lines 353-375). Specifically, regarding *IPMK* - missense and loss-of-function variants combine together to increase risk of AU, and *IPMK* promotes Toll-like receptor-induced inflammation by stabilizing *TRAF6*. This might be a valid mechanism affecting AU pathology that will need to be tested further. For *IDO2* - risk is observed with a clear loss-of-function of the gene. *IDO2* was reported necessary for the differentiation of regulatory T cells in vitro and has been shown to play a pro-inflammatory role in the development of B cell-mediated autoimmune arthritis. We hypothesize that the loss of *IDO2* might disrupt T-cell regulation and affect the T-Cell mediated response in the anterior chamber, contributing to the patho-mechanism of AU. We also examined tissue specific expression of both genes and validate that both are expressed in the iris and ciliary body. Regarding *STXBP2*, we have added mention of a known association between hemophagocytic lymphohistiocytosis, an inflammatory syndrome caused by variants in *STXBP2*, and uveitis (p.5; lines 122-128). Future directions for our group include further investigation into each of these targets, which will help us clarify any functional role they might have in AU.

- Some further discussion on the implications of the distinction between HLA-B*27 positive and negative AU would be useful, including how these findings may be translated in the future.

- This question is highly relevant in considering translational implications of the findings presented here. While the clinical manifestations between B*27-positive AU and B*27-negative AU overlap, B*27-positive AU is typically characterized by more robust inflammation and is more likely to recur than B*27-negative AU ^{1,2}. Furthermore, the identified protection of *ERAP1*-Hap10, brings up the possibilities of *ERAP1*'s therapeutic potential in the management of AU, and is particularly relevant in B*27-pos AU where recurrent episodes of inflammation that are difficult to control with topical steroids, put patients at risk of vision-threatening complications. We have added additional discussion on these points on the distinction of B*27 pos/neg disease and translational implications to the discussion of the updated manuscript (p. p.11; lines 314-316 and p.14; lines 380-382).

- While this is a well written paper, it would may useful to further highlight the key strengths of the work in context of the broader readership of the journal.

- Thanks to the reviewer for pointing out this opportunity to increase the impact of our publication. We present work impactful to a general audience for several reasons. Large scale analysis of biobank and health system cohorts with electronic health record data enables the study of relatively uncommon diseases with strong genetic risk factors, and whole exome sequencing at scale allows the discovery of genes with rare coding variants that impact risk. Further, we include all ancestries present in the data in our primary analyses, and dissect the contributions of different ancestries to the genetic risk factors we identified (in part thanks to the reviewers' comments). We now open our Discussion with these points of significance and have revised the paragraph to improve readability and make these points of significance more strongly (p. 11; lines 305-312).

1 Power, W. J., Rodriguez, A., Pedroza-Seres, M. & Foster, C. S. Outcomes in anterior uveitis associated with the HLA-B27 haplotype. *Ophthalmology* **105**, 1646-1651, doi:10.1016/S0161-6420(98)99033-9 (1998).

2 Rothova, A., Suttorp-van Schulten, M. S., Frits Treffers, W. & Kijlstra, A. Causes and frequency of blindness in patients with intraocular inflammatory disease. *Br J Ophthalmol* **80**, 332-336, doi:10.1136/bjo.80.4.332 (1996).

REVIEWERS' COMMENTS

Reviewer #1 (Remarks to the Author):

The authors have reviewed and revised this manuscript accordingly and recommend publication now that these questions have been answered. I feel that this manuscript will be an excellent addition to the field.